ecology

null models, community assembly, Soricidae, tropical forest, biogeography

**Author for correspondence:**
Frederik Van de Perre
e-mail: frederik.vandeperre@uantwerpen.be

# Functional volumes, niche packing and species richness: biogeographic legacies in the Congo Basin

Frederik Van de Perre[1], Michael R. Willig[2], Steven J. Presley[2], Itoka Jean-Claude Mukinzi[3], Mbalitini Sylvestre Gambalemoke[3], Herwig Leirs[1] and Erik Verheyen[1,4]

[1]Evolutionary Ecology Group, University of Antwerp, 2610 Antwerp, Belgium
[2]Department of Ecology & Evolutionary Biology, Center for Environmental Sciences & Engineering, and Institute of the Environment, University of Connecticut, Storrs, CT 06269-4210, USA
[3]Centre de Surveillance de la Biodiversité, University of Kisangani, Kisangani, Democratic Republic of the Congo
[4]OD Taxonomy and Phylogeny, Royal Belgian Institute for Natural Sciences, 1000 Brussels, Belgium

 FVdP, 0000-0002-0692-2060; MRW, 0000-0001-6884-9957;
SJP, 0000-0002-5987-0735; HL, 0000-0002-7612-5024;
EV, 0000-0001-7157-1474

Understanding the determinants of species coexistence in complex and species-rich communities is a fundamental goal of ecology. Patterns of species coexistence depend on how biotic interactions and environmental filtering act over ecological and evolutionary time scales. Climatic fluctuations in lowland rainforests of the Congo Basin led to the number of vertebrate species being significantly lower in central compared with northern ecoregions of the Basin. We used null models to assess whether climatic variations affected the community assembly of shrews. A consistent limit to functional similarity of species was not related to species richness. Rather, species richness is constrained by environmental factors, and these constraints are stronger in the central lowland forests of the Congo Basin. By constraining species geographic distributions, historical effects of rainforest refugia arising from climatic fluctuations may affect contemporary species composition of local shrew communities. The Congo River represents a vicariance event that led to allopatric speciation of shrews and continues to represent a barrier to dispersal. Ultimately, the historical effects of this barrier have led to differences in the functional volume of shrew communities in northern and central ecoregions. We

suggest that the analyses of community assembly can be used to identify Holocene refugia in the Congo Basin.

# 1. Introduction

Understanding the determinants of species coexistence in complex and species-rich communities is a fundamental goal of ecology [1,2]. Competition theory predicts that if two species have identical niches, either one species will exclude the other, or selective pressures will eventually result in character displacement [3,4]. At the same time, the physical environment imposes ecological and evolutionary constraints that create an 'ecological filter' such that species with similar ecological requirements occur in similar environments [5]. Ultimately, patterns of species coexistence depend on how both biotic interactions and environmental filtering act over ecological and evolutionary time scales [6]. Alternatively, community assembly can be modelled as neutral processes in which species have equivalent per capita demographic rates [7]. To enhance mechanistic understanding of the processes driving community assembly, the patterns of species coexistence can be studied over gradients of productivity or species richness [8,9]. The theory of limiting similarity [10] predicts that functional trait volume, defined with respect to multidimensional trait space, must increase as the number of species increases. An alternative view proposes that higher richness is associated with the denser packing of species in niche space, which could occur as productivity increases, increasing the potential number of viable populations that could be sustained in the same niche volume. Increased species packing could arise through finer specialization or greater overlap in resource use. These models of niche packing and expansion are not mutually exclusive and may occur in concert [11].

Species richness 'anomalies' are powerful natural experiments that facilitate the comparison of communities that differ in species richness, but share similar climatic conditions [12]. Lowland rainforests of the Congo Basin can be considered a species richness anomaly [13–15], as the number of vertebrate species is significantly lower in the central part of the Basin (i.e. left bank of the Congo River) compared with the northern part of the Basin (i.e. right bank of the Congo River) even though current environmental conditions, in terms of climate and tree species composition, are similar on both sides of the Congo River [16–18]. Rainforests of the Congo Basin have undergone cycles of contraction and expansion over the past 10 Myr in response to climatic fluctuations [19–21]. During dry periods, lowland rainforests retracted into refugia, where forest-dependent animals persisted until wetter and warmer climates facilitated re-expansion into their former geographic ranges [19]. The extent of central Congolian lowland forests (CLF) today is smaller than that of northern lowland forests (NLF). Consequently, historical speciation rates were probably lower and extinction rates were probably higher in the CLF than in the NLF [15]. Speciation rates increase with area because larger areas are more likely to experience vicariance events that enhance allopatric speciation [22]. Moreover, larger areas support larger population sizes and larger numbers of populations, reducing the likelihood of stochastic extinction events or extinction due to environmental perturbations [23]. In addition, the Congo River represents a transcontinental barrier that has limited dispersal of terrestrial species between the CLF and NLF for the past 34 Myr [24]. Based on these considerations, differences in richness between regions are expected to be greater for forest-dependent taxa with low dispersal capacities [15].

We use shrews (Soricidae) as a model group to evaluate niche packing and expansion within the lowland forests of the Congo Basin. Shrews are obligate insectivores that forage on the ground, enhancing the likelihood that competition will be intense among syntopic soricids when resources are limiting. In contrast with the situation for many forest vertebrates in the Congo Basin, shrews are not subject to human hunting pressures [25]. Herein, we take advantage of the species richness anomaly in lowland forest of the Congo Basin to investigate the relationship between species richness and functional diversity. More specifically, we (i) quantify packing and filling of functional trait space in relation to species richness and (ii) evaluate differences in the aspects of taxonomic and functional diversity between ecoregions.

# 2. Methods

## 2.1. Data collection

We used data on shrew research from the region around Kisangani [26], located in the centre of the Democratic Republic of Congo. The data represent 36 sampling sites distributed within six localities in Tshopo Province (figure 1). Sampling localities are separated by the Congo River and its major tributaries (Tshopo, Lindi and Lomami Rivers). Each sampling site is considered to represent a shrew community; hence, our dataset contains 15 communities of the CLF and 21 communities of the NLF (table 1).

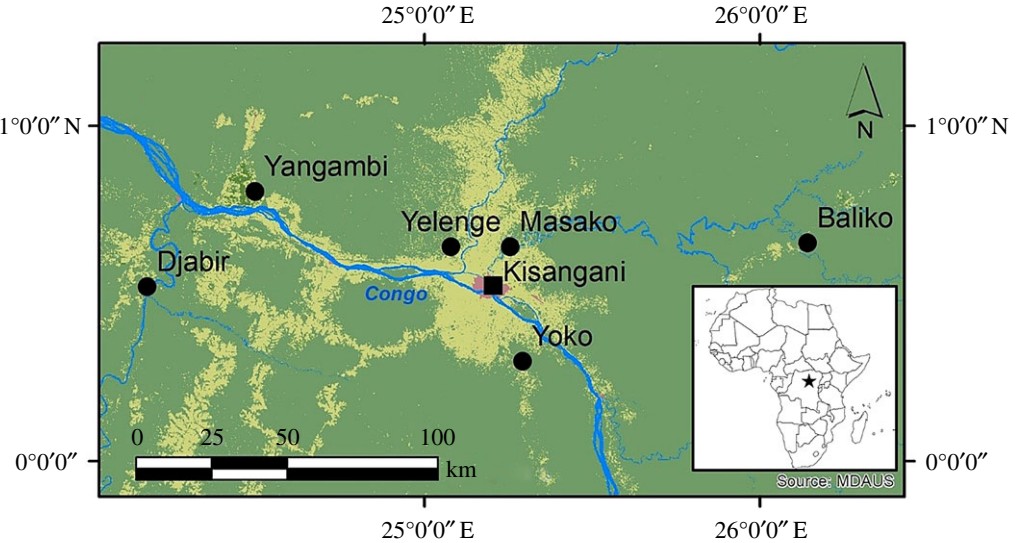

**Figure 1.** Sampling localities (dots) in the environs of Kisangani (see table 1 for additional details). The city of Kisangani (pink) is surrounded by a mosaic of agricultural land and regrowth forest (light green). Old-growth forests (dark green) can be found throughout the larger area. Blue represents the Congo River and its tributaries. The inset shows the location of the study area in Africa.

**Table 1** Number of sampling sites per ecoregion and per locality.

| locality | ecoregion | number of sampling sites |
| --- | --- | --- |
| Baliko | NLF | 3 |
| Djabir | CLF | 3 |
| Masako | NLF | 4 |
| Yangambi | NLF | 12 |
| Yelenge | NLF | 2 |
| Yoko | CLF | 12 |
| total | | 36 |

In all localities, shrews were sampled using the paceline method, which involved placing 20 pitfall traps at 5 m intervals on transects [27]. Pitfall traps consisted of non-baited buckets (30 cm deep) that were buried in the ground, with rims even with the ground surface. A plastic drift fence (100 m) was set to increase capture effectiveness by guiding shrews toward traps. Pitfall traps were maintained at their locations for 21 days and animals were sacrificed daily. Species were identified based on external morphology and cranio-dental characteristics. These species assignations were confirmed for several specimens of each species by molecular analysis (for more details, cf. [26]). Because the dataset of the Yoko locality contained multiple trapping sessions for each constituent site, we only used sessions that were restricted to a single season, thereby avoiding the use of data from transition periods between seasons. When multiple seasons remained for a particular site, we chose the trapping session with the highest total abundance of shrews as these would probably have most accurate measures of local diversity.

## 2.2. Functional trait data

Functional traits are phenotypic characteristics that influence ecosystem- or community-level processes [28]. Morphological attributes are informative for inferring ecological relationships among species at the community level; they correlate well with ecological characteristics and are good predictors of resource utilization [29,30]. Equally important, morphological measurements are highly repeatable and easily made compared with ecological characteristics (e.g. diet, behaviour). Although shrew species show little resource partitioning based on broad dietary categorizations of prey (i.e. all are

insectivores), variation in morphological traits can influence the efficiency with which each shrew species consumes different types of invertebrates. Because not all morphometric traits necessarily reflect relevant functional attributes of a species [28], we selected traits that characterize locomotion, foraging strategy and prey size.

Our selection of ecomorphological traits comprised external and cranial characteristics. External characteristics included mass (WT), length of body (HB), length of tail (T), length of hind foot (HF) and length of ear (E). Cranial characteristics included condylo-incisive length (CI), greatest width of skull (GW), interorbital width (IW), length of upper tooth row (UTR) and length of lower tooth row (LTR). Because most morphometric traits were strongly correlated with body size, and functional diversity metrics generally decrease as the strength of positive correlations between traits increases [31], we used relative measures to characterize function (i.e. T, HF and E were each divided by HB; GW, IW, UTR and LTR were each divided by the CI).

Outliers (i.e. observations beyond the $1.5 \times IQR$, where IQR, the interquartile range, is the difference between 75th and 25th percentiles) were removed from analyses, as these could be measurement or transcription errors, or in the case of mass, a consequence of pregnancy in females. The position of shrew species in two-dimensional trait space was visualized by non-metric multidimensional scaling (NMDS) using the metaMDS function of the R package vegan [32]. We used Gower distances (vegdist function in vegan) to represent pairwise functional distances between species.

## 2.3. Quantifying niche packing and filling of functional space

We quantified functional volume by calculating a multivariate convex hull volume, also known as functional richness (FRic; [33]). The convex hull volume is simply the smallest possible multidimensional volume that contains all species in a community. To estimate the packing of trait space, we calculated the mean nearest neighbour distance (MNND). Both measures were calculated in R [34] for each sampling site: we used the dbFD function of the FD package [35] to calculate FRic and the mean Gower distance from each species in the community to its nearest neighbour in multivariate trait space to determine MNND [36]. To explore differences among ecoregions in the relationships between FRic and species richness or between MNND and species richness, we used a fully factorial linear model (analysis of covariance (ANCOVA) based on type II sums of squares with ecoregion as a categorical factor and species richness as a covariate).

## 2.4. Null model analysis

To determine whether empirical values of FRic and MNND were significantly different from random expectation, we used null models. These models randomized species identities on the trait matrix 1000 times, and recalculated FRic and MNND for each community in each iteration. This generated a null distribution of 1000 values for each metric and each community. The empirical values and this null distribution were used to calculate a standardized effect size (SES) for each metric and each community. The SES was equal to the observed FRic or MNND minus the mean of the corresponding metric from the null distribution, divided by the standard deviation of the corresponding metric of the null distribution. Therefore, positive SESs indicated higher than expected FRic or MNND in an empirical community, while negative SESs indicated lower than expected values [36]. All calculations were performed in R.

Randomization of species identities via null models was based on two definitions of the species pool; one containing all species of the domain (i.e. the total species pool) and another in which each ecoregion was represented by its constituent species (i.e. regional species pools). This hierarchical approach facilitated the identification of processes that structure communities at different spatial and temporal scales [37]. The total pool contained all species recorded from lowland rainforests of the Congo Basin, whereas each of the regional species pools represented the set of species that have colonized and established within local communities of their respective ecoregions (CLF or NLF).

Statistical and conceptual problems arise when using an unconstrained null model to quantify MNND. Statistically, it is difficult to find a higher than expected MNND if an empirical volume is lower than expected given the total trait pool [36]. Conceptually, addressing questions related to the role of competitive interactions in shaping community structure requires consideration of whether species that can disperse to the focal community can also establish, given the local abiotic environment or habitat characteristics [37]. One way to avoid this issue is to assume that community assembly transpires in a hierarchical manner, where species first pass through a dispersal filter (i.e. be

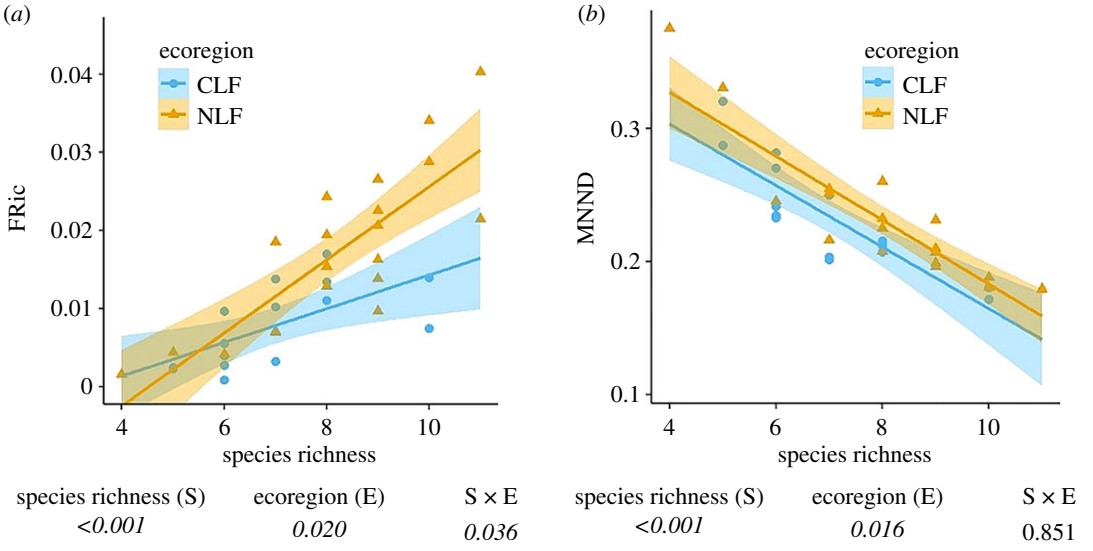

**Figure 2.** The relationship between functional richness (FRic) and species richness (a) and MNND and species richness (b) in shrew communities. Shaded areas represent 95% confidence intervals. The p-values are printed below each factor of the ANCOVA model. Italic p-values indicate significance.

able to reach the habitat) and then must deal with both abiotic and biotic filters represented by aspects of the environment. As such, trait volume may be reduced before abiotic responses or biotic interactions begin to operate due to dispersal limitation or historical contingency [36]. Consequently, a more appropriate null model for the MNND of a community constrains the species pool to species inside the empirical trait volume of the considered community, and from this constrained pool, the random communities can be generated to characterize a null distribution and standardized effect size value [36].

We used Wilcoxon signed-rank tests to determine whether SESs differed significantly from random expectations and a fully factorial linear model (ANCOVA using type II sums of squares) to determine whether SESs were related to species richness (covariate) and ecoregion (factor).

# 3. Results

## 3.1. Empirical multivariate functional richness and nearest neighbour distance

FRic increased with increasing species richness in each ecoregion. The slope of the relationship is significantly larger in the NLF compared with the CLF (i.e. significant interaction between species richness and ecoregion, the ANCOVA, figure 2a). MNND decreased with increasing species richness in each ecoregion. Slopes did not differ between ecoregions, but higher MNNDs were found in the NLF than in the CLF (figure 2b).

## 3.2. Null model analyses

Based on the total species pool, the SESs for FRic were significantly lower than expected in the CLF, but did not differ from random expectation in the NLF (figure 3a). Negative SESs indicate that community assembly is driven by environmental filtering as the empirical functional volume is less than expected given the empirical species richness [36]. The SESs for the MNND were significantly lower than expected in each ecoregion (figure 3a). Negative SESs indicate that the observed density of packing, determined by biotic interactions, is higher than expected given empirical species richness. Although the effects of environmental filtering and biotic interactions did not depend on species richness, they each differed significantly between ecoregions (lower SESs in CLF, table 2).

Based on the regional species pool, the SESs for functional richness were negative in both ecoregions (figure 3b). The SESs were not related to species richness or ecoregion (table 2). The SESs for MNND were significantly lower than expected in the CLF, but did not differ from random expectation in the NLF (figure 3b) and were not related to species richness or ecoregion (table 2).

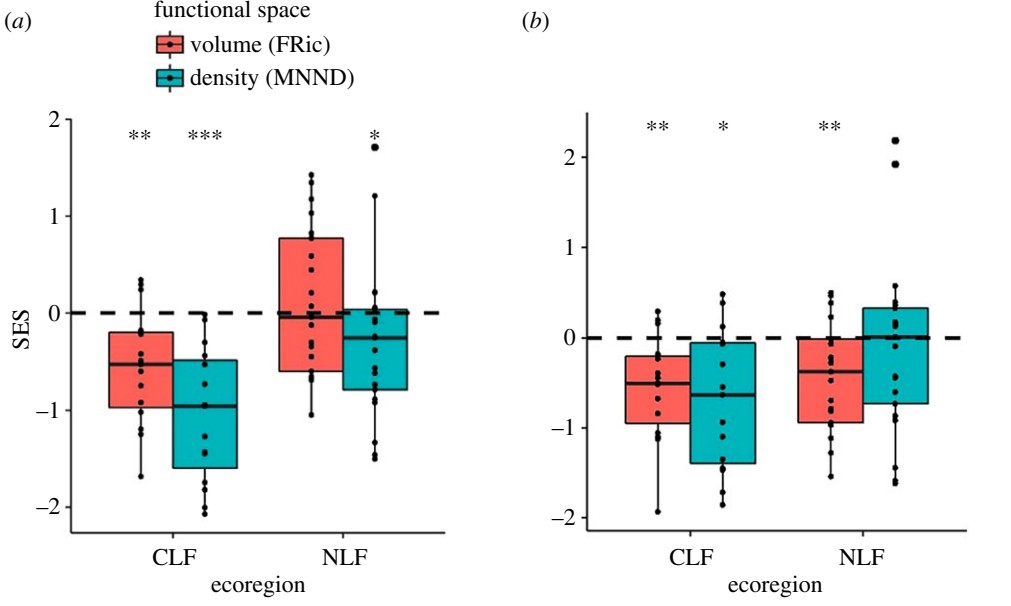

**Figure 3.** Standardized effect sizes (SESs) for FRic and MNND for shrew communities in the CLF and the NLF. The dashed lines indicate null expectations based on the total (*a*) or regional species pool (*b*) and the symbols indicate the significance level of Wilcoxon signed-rank tests (\*\*\* $p \leq 0.001$, \*\*$0.001 < p \leq 0.01$, \*$0.01 < p \leq 0.05$).

**Table 2.** The *p*-values of each factor in ANCOVA for each null model. Significant values are in italics.

| | total species pool | | | regional species pool | | |
|---|---|---|---|---|---|---|
| | species richness (S) | ecoregion (E) | S × E | species richness (S) | ecoregion (E) | S × E |
| unconstrained model | | | | | | |
| FRic | 0.352 | *0.021* | 0.123 | 0.548 | 0.398 | 0.197 |
| MNND | 0.632 | *0.015* | 0.593 | 0.304 | 0.140 | 0.290 |
| constrained model | | | | | | |
| MNND | *0.046* | 0.064 | 0.053 | 0.063 | 0.694 | 0.589 |

Based on the total species pool, SESs from the constrained randomizations were negative for the CLF, whereas those for the NLF did not differ significantly from random expectation (figure 4*a*). SESs did not differ between ecoregions, but were related to species richness (table 2). The SESs from the CLF communities were not correlated with empirical species richness, whereas those of the NLF were correlated with species richness (Pearson's *r*: −0.135 in CLF; 0.513 in NLF). The total species pool model is less appropriate for testing the effect of species interactions, as it is based on the unlikely assumption that all species are able to disperse to all communities. In the constrained model based on regional species pools, the empirical mean nearest neighbour distances in communities did not differ significantly from random expectation (figure 4*b*). The SESs from this null model were not related to species richness or ecoregion (table 2).

## 4. Discussion

As species richness increases, local communities can change in two ways: (i) by more dense packing of species into the niche space or (ii) by increasing the volume of the niche space to accommodate a larger number of species [30,38]. When species packing and niche volume are based on ecomorphological traits, the ecomorphological distances among species should decrease in the first case, whereas the volume of the morphological space occupied by all species should increase in the second case. For shrews in the Congo Basin, both functional volume and species packing increased with increasing empirical species

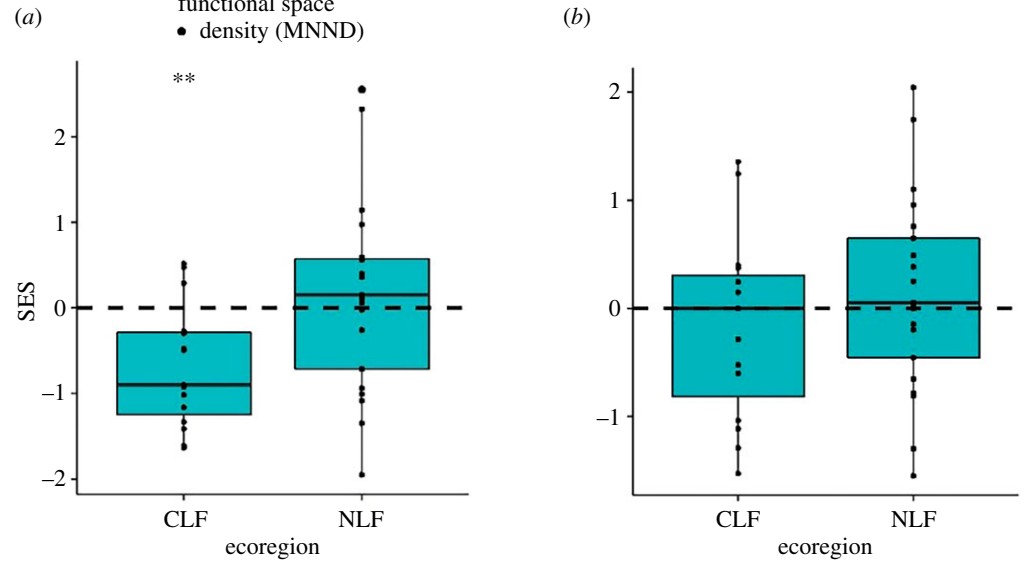

**Figure 4.** SESs for MNND of shrew communities in the CLF and NLF based on a constrained null model. The dashed lines indicate the null expectations based on the total (*a*) or regional (*b*) species pools, and symbols indicate the significance level of Wilcoxon signed-rank tests (*** $p \leq 0.001$, ** $0.001 < p \leq 0.01$, * $0.01 < p \leq 0.05$).

richness (figure 2). Consequently, we conducted analyses using null models to determine if increases in functional volume or decreases in MNND were different from that expected by chance. By controlling for variation in species richness among communities, we determined that increases in functional richness with increasing species richness were less than expected by chance, whereas reductions in MNND were consistent with random expectations. More specifically, the observed functional volume was generally smaller than that expected given species richness, as indicated by the negative standardized effect sizes for unconstrained null models (figure 3), whereas functional density did not differ from random expectation based on the constrained null model (figure 4). Furthermore, null models confirmed that the increase in functional richness was lower for CLF communities compared with NLF communities. This suggests that functional richness probably is constrained by environmental factors, and these constraints probably are stronger in the CLF than in the NLF.

Our first conclusion, that the species richness of shrew communities is not limited by species interactions contradicts theoretical predictions but agrees with patterns found in West-African shrews [39]. We identify three potential explanations for these contradictory findings. First, the apparent lack of niche partitioning could be due to essentially non-limiting resource availability (large diversity and abundance of terrestrial invertebrates; [40]) in tropical rainforests compared with local densities of shrew populations. Indeed, shrew communities in environments with limited availability of resources, such as the Siberian taiga [41,42] or tropical islands [43], show dietary niche partitioning whereas many shrew species in our study area are generalist and opportunistic feeders, with high dietary diversity and little dietary specialization [44]. Second, the absence of ecomorphological patterns consistent with competition theory for shrews in the Congo Basin may arise because competition is not (yet) sufficiently strong to lead to local extirpation, or because competition may not be the dominant process affecting community assembly [29]. Third, it is unclear if species are more finely partitioning a constrained trait space in more species-rich communities because intraspecific trait distributions and interspecific overlaps in those distributions are not well known for soricids [12]. Churchfield [45] noted that niches expand when few soricid species are syntopic, but that niche overlap decreases with the addition of species. Furthermore, competitive interactions can result in a shift in the distribution of abundances among species in alignment with their proximity in functional trait space [46]. Changes in species density in response to interspecific interactions have been revealed using density compensation analyses [47].

Our second conclusion is that functional diversity of shrew communities in lowland forests of the Congo Basin may be limited by environmental conditions. Environmental filtering postulates the exclusion of species with traits that do not represent effective adaptations to local conditions, yielding communities with more similar species than expected, given the pool of taxa available to colonize a site [48]. Based on the null model using the total species pool, environmental filtering is stronger in

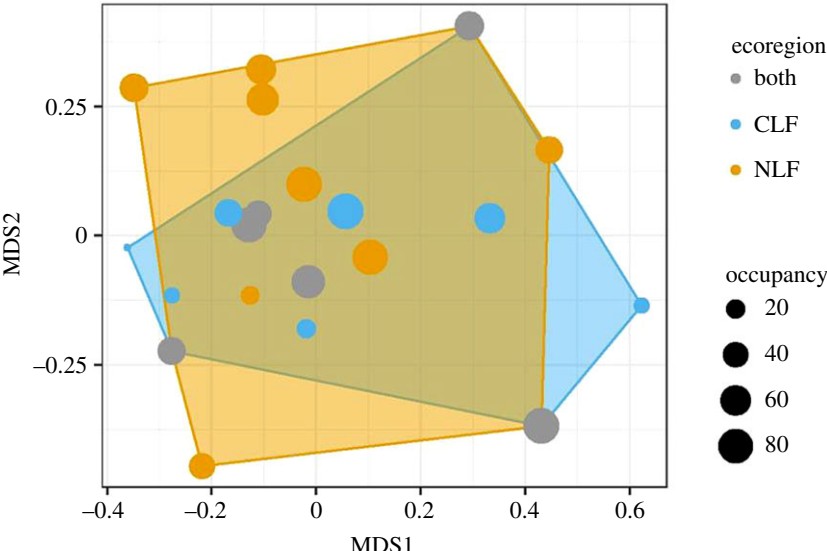

**Figure 5.** Position of shrew species in ecomorphological trait space. Information from ecomorphological traits was collapsed into two dimensions (MDS1 and MDS2) using NMDS. Figure S1b shows how the ecomorphological traits relate to these dimensions. Point colours indicate whether species occur only in the CLF (blue), only in the NLF (orange), or in both ecoregions (grey). Polygons represent two-dimensional Hull curves for the species of each ecoregion (orange for CLF, blue for NLF). Point size corresponds to the percentage of plots occupied by a species (i.e. occupancy).

the CLF compared with the NLF. This means that the species of the NLF are more peripheral (i.e. further from the centre) in functional space and therefore occupy a larger functional volume than do the species of the CLF. Indeed, when comparing the functional volumes of both ecoregions, four species occurring only within the NLF occur outside of the functional volume of the CLF, whereas only two species occurring within the CLF occur outside of the functional volume of the NLF (figure 5). Furthermore, the peripheral species are relatively common in the NLF, whereas the peripheral species are rare in the CLF. When constructing random communities from the total species pool, the probability is high that species that are outside the empirical functional volume of CLF communities will appear in randomized communities, and thus the SESs of such empirical communities will be negative.

The Congo River acts as a distributional barrier for many vertebrates [13,24,49–53] including shrews [54], and this geographic boundary may be responsible for maintaining differences in functional space between these ecoregions. Because the CLF is bounded by the Congo River in the north, east and west, and by savannah in the south, extinctions due to historical habitat fragmentation were unlikely to be compensated by subsequent dispersal to the same extent as in the NLF [15]. Therefore, the lack of peripheral species in the CLF may be related to their limited dispersal capacities, particularly with respect to large rivers. Indeed, *Sylvisorex johnstoni* (a peripheral species; electronic supplementary material, figure S1) and *S. ollula* (a species that occurs in both ecoregions) responded differently to palaeo-environmental changes in the western Congo Basin [55]. Haplotype analyses showed that relict populations of *S. johnstoni* that survived in forest fragments during arid periods did not increase in geographic range size when forest re-expanded, whereas *S. ollula* rapidly (re)colonized a large part of the area [55]. The different responses to palaeo-environmental changes can be linked to higher extinction rates and lower dispersal capacities of *S. johnstoni*, which in turn could be related to its small body size compared with that of *S. ollula*. This provides a basis for understanding why three of the smallest species in the NLF are peripheral in ecomorphological space (electronic supplementary material, figure S1). Note that differences in geographic extent and sampling effort between ecoregions can affect the richness of the regional species pools. Because of the lower number of sampling sites and individuals captured in the CLF (15 sites and 523 individuals) compared with the NLF (21 sites and 750 individuals), a potential bias could result in an underestimation of the richness and functional volume of the CLF species pool compared with that in the NLF. Despite this potential bias, the NLF species pool contained only one more species than the CLF species pool. *A priori*, we expected the NLF species pool to be more species rich compared with that of the CLF due to the empirical species richness anomaly associated with historical biogeography [13–15]. Consequently, we have little evidence that

unequal sampling efforts or sample sizes between the two regions created a bias that would affect the conclusions of this study. Importantly, only analyses based on the total species pool included data that could bias results, as other analyses used a community as the focal unit or were restricted to a single region.

Aside from limited dispersal between ecoregions, models based on regional pools indicate that environmental filtering is taking place within each ecoregion. The environmental filtering in models based on regional pools cannot be attributed to the barrier effect of the Congo River, as the models only contain species that have colonized a community within an ecoregion. Rather, peripheral species are less common than expected in local communities because they have difficulties surviving in those habitats, because they have difficulties reaching those habitats, or because the resources on which they specialize are rare in those habitats. Species' distance to the centre of functional space is negatively correlated to their average abundance (Spearman correlation: $\rho = -0.451$, $p = 0.040$). Therefore, those species that enlarge a community's functional space also have highest probabilities of becoming locally extinct. Soricids are not severely affected by small-scale disturbances as a multi-taxon study including vertebrates, invertebrates, fungi and plants found that shrews were the only group whose richness, diversity and species composition did not differ among forests of different carbon storage [56]. However, large-scale forest disturbances such as those provoked by past climatic fluctuations in the Congo Basin can affect shrew species composition [57]. During arid periods, dense forest was restricted to large refugia or isolated forest fragments within the forest–savannah matrix. Shrew species are more likely to survive in the forest–savannah matrix outside of dense forest refugia if they are able to prey on a large range of invertebrates. Consequently, generalists, species that are close to the centre in ecomorphological space, are likely to be dominant, whereas more functionally peripheral species might exhibit low abundances or become locally extinct. Furthermore, as generalist species were already established when forests re-expanded they had a competitive advantage as opposed to the reinvading dense forest species [58].

Our results suggest that the biogeographic history of our study area may have left a distinctive imprint on the composition and diversity of shrew communities. Our sampling localities are probably outside of major Holocene refugia and therefore gradually became moist rainforest about 2000 years ago [21]. Because of the low dispersal abilities and small home range sizes of shrews [54], and the barrier effect exerted by tributaries of the Congo River [59], the effect of these historical extirpations are reflected in the composition of local communities (i.e. legacy effects [60]). Indeed, species (i.e. *Crocidura ludia, C. littoralis, C. denti, C. olivieri*) described by Barrière *et al.* [57] as mainly occurring in transition zones, such as secondary forest or forest–savannah mosaics, are among the most abundant species in our study area, in which dense forest species are more rare. The fact that these species are still dominant can be explained by niche pre-emption: species that survived in forest–savannah reduced the amount of resources available to other species and, in doing so, limited the local abundance that could be attained by reinvading species that need these resources to survive and reproduce [61]. This historical contingency thus explains why we found no evidence for limiting similarity. The difference in functional volume between ecoregions is consistent with the contemporary and historical effects of the barrier represented by the Congo River, which acts as a filter for species with low dispersal capacities. A limitation in our study is that species pools were constructed from samples of local communities rather than from independent data. Some rare species may have been overlooked, which would result in an underestimation of the richness and ecomorphological volume of the species pool [62]. If the positions of overlooked species based on functional characteristics are outside of the estimated functional space based on sampled communities, the effects of habitat filtering would be underestimated. Conversely, if the functional characteristics of overlooked species occur within the functional volume based on the sampled communities, mean nearest neighbour distances would be overestimated in the species pool, resulting in underestimation of the effects of interspecific competition. Because peripheral species are rarest, we expect that habitat filtering rather than species interactions is more likely to be underestimated. Analyses of community assembly, such as ours, can identify major Holocene refugia within the Congo Basin: null analyses based on the regional species pool of shrew communities in former Holocene refugia are likely to show patterns opposite to those found in our study area. Shrew communities in forests unaffected by past disturbances will have no priority effects and will therefore probably show evidence for limiting similarity.

Data accessibility. All data are available at http://projects.biodiversity.be/africanmammalia.

Authors' contributions. F.V.d.P., M.R.W. and S.J.P. analysed the data. F.V.d.P., M.R.W., S.J.P and E.V. wrote the manuscript, after which all authors reviewed the text and interpretations.

Competing interests. The authors declare that no competing interests exist.

Acknowledgements. This study is a deliverable of the COBIMFO Project (Congo Basin integrated monitoring for forest carbon mitigation and biodiversity; contract no. SD/AR/01A) and was funded by the Belgian Science Policy Office

(Belspo). F.V.d.P. was supported by a PhD fellowship from the Research Foundation–Flanders. M.R.W. and S.J.P. were supported by the Center for Environmental Sciences and Engineering, Institute of the Environment, at the University of Connecticut, as well as by NSF grant nos. DEB-1239764 and DEB-1546686.

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
