## [Reviewer comments · Royal Society Open Science]

Review History

RSOS-191582.R0 (Original submission)

Review form: Reviewer 1

Is the manuscript scientifically sound in its present form?

No

Are the interpretations and conclusions justified by the results?

No

Is the language acceptable?

Yes

Do you have any ethical concerns with this paper?

No

Have you any concerns about statistical analyses in this paper?

No

Recommendation?

Accept with minor revision (please list in comments)

Comments to the Author(s)

The manuscript presents an indirect assessment of community assembly processes based on morphometric analyses of shrews in the Congo Basin.

The writing is clear and concise; the methods are sound; and the interpretations are congruent with the results. I have two concerns (see two paragraphs below), each of which the authors should be able to address.

The authors claim to include a table containing “additional details” (see Fig. 1 caption) regarding sampling localities. However, I could not find that table, neither in the main file nor in the file containing Supplementary Info. Without those details, it is difficult to determine the distribution of sampling effort between CLF and NLF. Since 2 of the 6 dots in Fig. 1 are south of the Congo, whereas 4 are north, I suspect that the sampling was uneven. In addition, from the map it looks like the geographic extent of sampling localities was greater for NLF than CLF. If so, it would be useful for the authors to explicitly address the concern that differences in sampling intensity and broader geographic (less spatially restricted) sampling in NLF could themselves reflect as differences in shrew community structure.

The authors base the justification for a particular null model on a claim that “community assembly transpires in a hierarchical manner, where species first pass through an abiotic filter and then a biotic filter...” (pg. 7), which is incorrect. There is no necessary order to abiotic and biotic filtering in community assembly. For example, imagine a shrew species that arrives to a new site during the wet season, but cannot survive there owing to an endemic disease organism, so its population fails to persist, even though it would also have failed during the dry season that presents conditions outside of the shrew’s fundamental niche, had it persisted through the wet season to the dry. The logic in this paragraph needs to be fixed.

Line 89 – edit “we only used sessions were restricted to” to improve clarity.

Line 176 – replace “indicated” with “indicate”.

Review form: Reviewer 2**Is the manuscript scientifically sound in its present form?**

No

Are the interpretations and conclusions justified by the results?

No

Is the language acceptable?

No

Do you have any ethical concerns with this paper?

No

Have you any concerns about statistical analyses in this paper?

Yes

Recommendation?

Major revision is needed (please make suggestions in comments)

Comments to the Author(s)

Comments to Authors:

This is an interesting study on a fascinating yet under-explored fauna and region of the world. I commend the authors for gathering all of these data, however the location of archived samples and the methods used to identify these poorly studied species need to be addressed. I take some issues with the interpretation of the results, and the strength of the statements in the discussion. The largest concern is the discrepancy between the results that say community ecological volume is increasing with species richness, and the interpretation that “species became more tightly packed in the functional volume rather than expanding the size of the functional volume.” The results seem to be in sharp contrast to this statement. Also, the strength of the conclusion that environment is the cause of the differences between communities is less than the authors make it out to be. Environmental filtering is one of the alternative explanations, but not the definitive reason.

I would also like to see more details in the methods.

I hope my comments can be useful to the authors.

Abstract:

Lines 23-24: Confusing statement about central and north, perhaps “number of species is lower in the central ecoregion than the northern ecoregion”.

Line 25: “...these climatic variations...” Which climatic variations?

Line 27: Environmental factors are not “stronger” in some places than others. Their effects on species richness may be.

Line 28-30: Sentence that begins “An effect of environmental filtering...” is confusing. It can be hard to fit so much into a limited abstract, but this sentence needs to be re-worded to be understood.

Introduction:

Lines 43-44: I’m not sure about “ecological time-scales”, just using evolutionary time scale here may be more clear.

Lines 44-45: I don’t know if Hubble said that “community assembly can be a largely neutral process”. I think he modeled assembly as a neutral process and found some theoretical (and a little bit of empirical) support for this model.

Line 45: Enhance mechanistic understanding of what?

Line 46-51. It is interesting that both hypotheses, niche expansion and niche packing, could both be driven by specialization. I am not saying that this is what the references intended to say, but it makes me think that limiting similarity and increased packing by finer specialization could be two sides to the same idea, however niche expansion, I am guessing, would require specialization on a resource not previously in the niche volume, and niche packing would require specialization WITHIN the volume, though that may be hard to disentangle. It could be useful to clarify these ideas here and perhaps later in the manuscript.

Methods: In general, more information is needed in the methods. One of the biggest things that is missing is how species were delimited. I believe that shrews can be very difficult to tell apart, and I’m sure the shrews of the DRC are likely understudied, and these regions likely include cryptic species. Are these specimens archived? Do add some information of how the authors went from shrews in a bucket to samples with accurate species identifications.

Line 85: Tell us more about where Kisangani is. So far, the authors have only mentioned the Congo Basin, not the DRC.

Line 88-90: Wouldn't trapping in multiple seasons potentially yield more species? Since the reader doesn't know how you are going to partition your data for analysis, this information on Yoko is confusing here. It may be better to put it in the next section.

Line 104: This is not a complete sentence. Reword please.

Line 108-110: Please give more information of why you did this "size correction" and cite some sources.

Line 114-115: Please state what vegan command you used, number of axes, number of iterations, etc. Also, why was NMDS used rather than some other ordination method? Correct me if I'm wrong, but I don't think RSOS has strict page limits. An extra sentence or two explaining your choice of methods will go a long way in clarifying your results to the reader.

Line 117: Cite R package, and give command and relevant settings.

Line 118: Was this convex hull calculated from raw variables, or NMDS scores that were generated in the previous section? Or was some other ordination method used here? This is very important. How was the data partitioned and how many convex hulls are there? 2, one per "region", or 1 per community (6)?

Line 119-120: How was MNND and Gower distance calculated?

Line 128-129: I thought that the "regional species pool" method is what was done in the previous section. It is not clear how the null is different from the alternative. See method comments above.

Line 131: Careful with "adapted" here. Isn't a neutral theory being tested? The total pool just contains all the species in the lowland rainforests.

Line 134: remove "regardless"

Lines 135-137: Explain how you calculated this (which R(?) package), same as above.

Lines 142-152: I'm a little confused by this paragraph. This sounds like a justification of the methods that you used in the above paragraph. As far as I can tell, your null models did constrain the traits using empirical trait volume. If I am reading this right, and I may not be, I recommend moving this paragraph to the start of the null model methods section so readers understand your choice of null model evaluation prior to describing the exact methods.

Results:

Line 160: Provide some numbers here, and explain "significantly larger slope" (i.e. effect)

Line 161: Are these slopes "significantly different" as in line 160? This will affect your statement here.

Line 167-169: Citation needed.

Discussion:

Line 202-203: I think you mean "If ecomorphological traits can effectively delimit niche occupancy,..." I don't think species packing is determined by ecomorphology, and, since you estimated niche volume FROM ecomorphological traits, then it is inevitable that your volumes are based on these traits.

Lines 205-207: I am confused by this interpretation. It seems that functional volume increases with species richness (Fig. 1A). This is a volume-based metric calculated with the FRic method. If FRic increases with species richness, this means that ecomorphological volume is increasing, and therefore this supports the hypothesis of limiting similarity and NOT the hypothesis of niche packing in the same volume. This is the exact opposite of what is said on line 206-207, where you say that functional volume DOES NOT increase with species richness, despite reporting that it does.

Line 208-210: I do not know how environmental factors come into play here, as they, as far as I can tell from the methods, were never tested. You describe the history of the region in the introduction, and then state that the environments north and south of the river are very similar. What I think you mean is that historical factors MAY explain the differences in the communities, not environmental factors DO explain.

Line 214: Need to describe what "limited" means here. Limited in richness? In morphological disparity?

Line 230: I don't know if they are "likely limited" as this is just one of many alternative hypotheses. "May be limited..." seems more appropriate.

Lines 235-240: These results are consistent with the hypothesis of limiting similarity. See comments above. Community ecological volume increases with species richness (Figure 1A), and the rare species in the CLF are outside the central morphospace. Figure 5 is a nice example of this. However, aren't the axes and distances in NMDS meaningless? As in the distances represented in NMDS are not "true" distances? A PCA/PCoA plot of this would be interesting to see as well, but maybe not required.

Line 267-269: ...or because the resources that they are specialized in are less abundant in the habitat, or they are harder to catch in pitfalls.

Line 285: This is very interesting, but I don't think your results make this definitive. "Our results suggest that the biogeographic history of the basin may have left an imprint on...."

Line 289-290: Isn't the expectation that specialists are in lower numbers than "generalists" found in the center of morphospace? Also I don't think "enjoy" is the best word here. We don't really know what, or if, shrews enjoy.

Line 292: consistent with, not necessarily "caused by"

Line 326: Is the editor of this volume the same as the second co-author? The names are spelled differently

Review form: Reviewer 3

Is the manuscript scientifically sound in its present form?

Yes

Are the interpretations and conclusions justified by the results?

Yes

Is the language acceptable?

Yes

Do you have any ethical concerns with this paper?

No

Have you any concerns about statistical analyses in this paper?

No

Recommendation?

Accept with minor revision (please list in comments)

Comments to the Author(s)

RSOS_191582

Title: Functional volumes, niche packing, and species richness: biogeographic legacies in the Congo Basin

Author: Frederik van de Perre

General Comments

The paper examines coexistence of shrew species in parts of the Congo Basin to explain why one portion of the basin harbors more species than a different portion. The approach is to examine niche space of the different species.

The study addresses an important question in species diversity and uses a somewhat interesting approach.

The paper is quite readable. My comments pertain to a some of the assumptions, and whether the findings are conclusive or not. Mostly they require clarification for a fairly naïve reader.

Specific Comments

These are in no particular order.

1) I am not a shrew biologist, and thus I was a bit surprised to learn, in the discussion, section that all of the species are generalist feeders. Therefore, diet plays no part in niche. This is fine. However, it should have been mentioned in the introduction. Indeed, after reading the discussion, it is not clear to me how you concluded that (line 76) 'competition will be intense'. It seems to me that generalists can switch prey items easily (?).

2) To me, niche is more than ecomorphological traits. The morphological aspect is clear. However, it would help readers to expand a bit on the ecological aspect. Again, reading the paper alone, it is not clear to me how these species separate among themselves ecologically.

3) I assume that quantifying niche packing and filling of functional space statistically is accepted. However, it might help readers to describe the extent of resources in the two portions of the study area. Perhaps, I am too wedded to Geographical Ecology, by Robert MacArthur, by niche packing and filling to me is the broken-stick model and length of the stick. I did not get a feel for available resources by 'multivariate convex hull volume'.

4) The Results section reads well. But there seems to be a large leap of logic from the Results to the last paragraph of the Discussion, stressing the distinctive imprint of Holocene climate. I assume the imprint is in the species composition and ecomorphological traits, by I suspect that naïve readers will miss the connection, too.

Sorry if these comments are naïve. But the paper should be understandable to a non-shrew, non-Congo Basin ecologist.

Decision letter (RSOS-191582.R0)

27-Nov-2019

Dear Mr Van de Perre,

The editors assigned to your paper ("Functional volumes, niche packing, and species richness: biogeographical legacies in the Congo Basin") have now received comments from reviewers. We would like you to revise your paper in accordance with the referee and Associate Editor suggestions which can be found below (not including confidential reports to the Editor). Please note this decision does not guarantee eventual acceptance.

Please submit a copy of your revised paper before 20-Dec-2019. Please note that the revision deadline will expire at 00.00am on this date. If we do not hear from you within this time then it will be assumed that the paper has been withdrawn. In exceptional circumstances, extensions may be possible if agreed with the Editorial Office in advance. We do not allow multiple rounds of revision so we urge you to make every effort to fully address all of the comments at this stage. If deemed necessary by the Editors, your manuscript will be sent back to one or more of the original reviewers for assessment. If the original reviewers are not available, we may invite new reviewers.

- Data accessibility

It is a condition of publication that all supporting data are made available either as supplementary information or preferably in a suitable permanent repository. The data accessibility section should state where the article's supporting data can be accessed. This section should also include details, where possible of where to access other relevant research materials such as statistical tools, protocols, software etc can be accessed. If the data have been deposited in

an external repository this section should list the database, accession number and link to the DOI for all data from the article that have been made publicly available. Data sets that have been deposited in an external repository and have a DOI should also be appropriately cited in the manuscript and included in the reference list.

<http://datadryad.org/submit?journalID=RSOS&manu=RSOS-191582>

- **Competing interests**

- **Authors' contributions**

- **Acknowledgements**

- **Funding statement**

Best regards,

on behalf of the Associate Editor, and Professor Kevin Padian (Subject Editor)
openscience@royalsociety.org

Associate Editor's comments to Author:

We're recommending a revision: though the reviewers are broadly positive towards your work, a number of items need careful attention before we can move it forward. Please ensure you

thoroughly address the reviewers' concerns, and bear in mind that further review may be required upon submission of the revision, so providing a marked-up version of the revision (alongside a point-by-point response) will help the editors and reviewers.

Reviewers' Comments to Author:

Reviewer: 1

Comments to the Author(s)

The manuscript presents an indirect assessment of community assembly processes based on morphometric analyses of shrews in the Congo Basin.

The writing is clear and concise; the methods are sound; and the interpretations are congruent with the results. I have two concerns (see two paragraphs below), each of which the authors should be able to address.

The authors claim to include a table containing "additional details" (see Fig. 1 caption) regarding sampling localities. However, I could not find that table, neither in the main file nor in the file containing Supplementary Info. Without those details, it is difficult to determine the distribution of sampling effort between CLF and NLF. Since 2 of the 6 dots in Fig. 1 are south of the Congo, whereas 4 are north, I suspect that the sampling was uneven. In addition, from the map it looks like the geographic extent of sampling localities was greater for NLF than CLF. If so, it would be useful for the authors to explicitly address the concern that differences in sampling intensity and broader geographic (less spatially restricted) sampling in NLF could themselves reflect as differences in shrew community structure.

The authors base the justification for a particular null model on a claim that "community assembly transpires in a hierarchical manner, where species first pass through an abiotic filter and then a biotic filter..." (pg. 7), which is incorrect. There is no necessary order to abiotic and biotic filtering in community assembly. For example, imagine a shrew species that arrives to a new site during the wet season, but cannot survive there owing to an endemic disease organism, so its population fails to persist, even though it would also have failed during the dry season that presents conditions outside of the shrew's fundamental niche, had it persisted through the wet season to the dry. The logic in this paragraph needs to be fixed.

Line 89 – edit "we only used sessions were restricted to" to improve clarity.

Line 176 – replace "indicated" with "indicate".

Reviewer: 2

Comments to the Author(s)

Comments to Authors:

This is an interesting study on a fascinating yet under-explored fauna and region of the world. I commend the authors for gathering all of these data, however the location of archived samples and the methods used to identify these poorly studied species need to be addressed. I take some issues with the interpretation of the results, and the strength of the statements in the discussion. The largest concern is the discrepancy between the results that say community ecological volume is increasing with species richness, and the interpretation that "species became more tightly packed in the functional volume rather than expanding the size of the functional volume." The results seem to be in sharp contrast to this statement. Also, the strength of the conclusion that environment is the cause of the differences between communities is less than the authors make it

out to be. Environmental filtering is one of the alternative explanations, but not the definitive reason.

I would also like to see more details in the methods.

I hope my comments can be useful to the authors.

Abstract:

Lines 23-24: Confusing statement about central and north, perhaps “number of species is lower in the central ecoregion than the northern ecoregion”.

Line 25: “...these climatic variations...” Which climatic variations?

Line 27: Environmental factors are not “stronger” in some places than others. Their effects on species richness may be.

Line 28-30: Sentence that begins “An effect of environmental filtering...” is confusing. It can be hard to fit so much into a limited abstract, but this sentence needs to be re-worded to be understood.

Introduction:

Lines 43-44: I’m not sure about “ecological time-scales”, just using evolutionary time scale here may be more clear.

Lines 44-45: I don’t know if Hubble said that “community assembly can be a largely neutral process”. I think he modeled assembly as a neutral process and found some theoretical (and a little bit of empirical) support for this model.

Line 45: Enhance mechanistic understanding of what?

Line 46-51. It is interesting that both hypotheses, niche expansion and niche packing, could both be driven by specialization. I am not saying that this is what the references intended to say, but it makes me think that limiting similarity and increased packing by finer specialization could be two sides to the same idea, however niche expansion, I am guessing, would require specialization on a resource not previously in the niche volume, and niche packing would require specialization WITHIN the volume, though that may be hard to disentangle. It could be useful to clarify these ideas here and perhaps later in the manuscript.

Methods: In general, more information is needed in the methods. One of the biggest things that is missing is how species were delimited. I believe that shrews can be very difficult to tell apart, and I’m sure the shrews of the DRC are likely understudied, and these regions likely include cryptic species. Are these specimens archived? Do add some information of how the authors went from shrews in a bucket to samples with accurate species identifications.

Line 85: Tell us more about where Kisangani is. So far, the authors have only mentioned the Congo Basin, not the DRC.

Line 88-90: Wouldn’t trapping in multiple seasons potentially yield more species? Since the reader doesn’t know how you are going to partition your data for analysis, this information on Yoko is confusing here. It may be better to put it in the next section.

Line 104: This is not a complete sentence. Reword please.

Line 108-110: Please give more information of why you did this “size correction” and cite some sources.

Line 114-115: Please state what vegan command you used, number of axes, number of iterations, etc. Also, why was NMDS used rather than some other ordination method? Correct me if I'm wrong, but I don't think RSOS has strict page limits. An extra sentence or two explaining your choice of methods will go a long way in clarifying your results to the reader.

Line 117: Cite R package, and give command and relevant settings.

Line 118: Was this convex hull calculated from raw variables, or NMDS scores that were generated in the previous section? Or was some other ordination method used here? This is very important. How was the data partitioned and how many convex hulls are there? 2, one per "region", or 1 per community (6)?

Line 119-120: How was MNND and Gower distance calculated?

Line 128-129: I thought that the "regional species pool" method is what was done in the previous section. It is not clear how the null is different from the alternative. See method comments above.

Line 131: Careful with "adapted" here. Isn't a neutral theory being tested? The total pool just contains all the species in the lowland rainforests.

Line 134: remove "regardless"

Lines 135-137: Explain how you calculated this (which R(?) package), same as above.

Lines 142-152: I'm a little confused by this paragraph. This sounds like a justification of the methods that you used in the above paragraph. As far as I can tell, your null models did constrain the traits using empirical trait volume. If I am reading this right, and I may not be, I recommend moving this paragraph to the start of the null model methods section so readers understand your choice of null model evaluation prior to describing the exact methods.

Results:

Line 160: Provide some numbers here, and explain "significantly larger slope" (i.e. effect)

Line 161: Are these slopes "significantly different" as in line 160? This will affect your statement here.

Line 167-169: Citation needed.

Discussion:

Line 202-203: I think you mean "If ecomorphological traits can effectively delimit niche occupancy,..." I don't think species packing is determined by ecomorphology, and, since you estimated niche volume FROM ecomorphological traits, then it is inevitable that your volumes are based on these traits.

Lines 205-207: I am confused by this interpretation. It seems that functional volume increases with species richness (Fig. 1A). This is a volume-based metric calculated with the FRic method. If FRic increases with species richness, this means that ecomorphological volume is increasing, and therefore this supports the hypothesis of limiting similarity and NOT the hypothesis of niche packing in the same volume. This is the exact opposite of what is said on line 206-207, where you say that functional volume DOES NOT increase with species richness, despite reporting that it does.

Line 208-210: I do not know how environmental factors come into play here, as they, as far as I can tell from the methods, were never tested. You describe the history of the region in the introduction, and then state that the environments north and south of the river are very similar.

What I think you mean is that historical factors MAY explain the differences in the communities, not environmental factors DO explain.

Line 214: Need to describe what “limited” means here. Limited in richness? In morphological disparity?

Line 230: I don’t know if they are “likely limited” as this is just one of many alternative hypotheses. “May be limited...” seems more appropriate.

Lines 235-240: These results are consistent with the hypothesis of limiting similarity. See comments above. Community ecological volume increases with species richness (Figure 1A), and the rare species in the CLF are outside the central morphospace. Figure 5 is a nice example of this. However, aren’t the axes and distances in NMDS meaningless? As in the distances represented in NMDS are not “true” distances? A PCA/PCoA plot of this would be interesting to see as well, but maybe not required.

Line 267-269: ...or because the resources that they are specialized in are less abundant in the habitat, or they are harder to catch in pitfalls.

Line 285: This is very interesting, but I don’t think your results make this definitive. “Our results suggest that the biogeographic history of the basin may have left an imprint on....”

Line 289-290: Isn’t the expectation that specialists are in lower numbers than “generalists” found in the center of morphospace? Also I don’t think “enjoy” is the best word here. We don’t really know what, or if, shrews enjoy.

Line 292: consistent with, not necessarily “caused by”

Line 326: Is the editor of this volume the same as the second co-author? The names are spelled differently

Reviewer: 3
Comments to the Author(s)

RSOS_191582

Title: Functional volumes, niche packing, and species richness: biogeographic legacies in the Congo Basin

Author: Frederik van de Perre

General Comments

The paper examines coexistence of shrew species in parts of the Congo Basin to explain why one portion of the basin harbors more species than a different portion. The approach is to examine niche space of the different species.

The study addresses an important question in species diversity and uses a somewhat interesting approach.

The paper is quite readable. My comments pertain to a some of the assumptions, and whether the findings are conclusive or not. Mostly they require clarification for a fairly naïve reader.

Specific Comments

These are in no particular order.

1) I am not a shrew biologist, and thus I was a bit surprised to learn, in the discussion, section that all of the species are generalist feeders. Therefore, diet plays no part in niche. This is fine. However, it should have been mentioned in the introduction. Indeed, after reading the discussion, it is not clear to me how you concluded that (line 76) 'competition will be intense'. It seems to me that generalists can switch prey items easily (?).

2) To me, niche is more than ecomorphological traits. The morphological aspect is clear. However, it would help readers to expand a bit on the ecological aspect. Again, reading the paper alone, it is not clear to me how these species separate among themselves ecologically.

3) I assume that quantifying niche packing and filling of functional space statistically is accepted. However, it might help readers to describe the extent of resources in the two portions of the study area. Perhaps, I am too wedded to Geographical Ecology, by Robert MacArthur, by niche packing and filling to me is the broken-stick model and length of the stick. I did not get a feel for available resources by 'multivariate convex hull volume'.

4) The Results section reads well. But there seems to be a large leap of logic from the Results to the last paragraph of the Discussion, stressing the distinctive imprint of Holocene climate. I assume the imprint is in the species composition and ecomorphological traits, by I suspect that naïve readers will miss the connection, too.

Sorry if these comments are naïve. But the paper should be understandable to a non-shrew, non-Congo Basin ecologist.

Author's Response to Decision Letter for (RSOS-191582.R0)

See Appendix A.

RSOS-191582.R1 (Revision)

Review form: Reviewer 2

Is the manuscript scientifically sound in its present form?

Yes

Are the interpretations and conclusions justified by the results?

Yes

Is the language acceptable?

Yes

Do you have any ethical concerns with this paper?

No

Have you any concerns about statistical analyses in this paper?

No

Recommendation?

Accept as is

Comments to the Author(s)

I believe the authors did a nice job at addressing my concerns with the manuscript in a thoughtful manner. I appreciate the clarity provided on some of the methods that I was confused on, and their willingness to make the appropriate changes. I think however that it is a good idea to post all relevant data and scripts in an easy to access format. I commend the authors for their cool and interesting piece of work and their contribution to African and mammalian evolutionary ecology.

Decision letter (RSOS-191582.R1)

22-Jan-2020

Dear Mr Van de Perre,

It is a pleasure to accept your manuscript entitled "Functional volumes, niche packing, and species richness: biogeographic legacies in the Congo Basin" in its current form for publication in Royal Society Open Science. The comments of the reviewer(s) who reviewed your manuscript are included at the foot of this letter.

Additionally, in regards to the final remaining comments from Reviewer 2, we wish to query whether you can either provide the relevant data and scripts associated with your manuscript as supplementary material (and please send us a zip file for us to upload to your submission), OR upload all relevant data and scripts to the Dryad Digital Repository through our integrated system, free of charge. Please see Reviewer 2's comments at the end of this email.

Once these relevant files have been received, you can expect to receive a proof of your article in the near future. Please contact the editorial office (openscience_proofs@royalsociety.org) and the production office (openscience@royalsociety.org) to let us know if you are likely to be away from e-mail contact -- if you are going to be away, please nominate a co-author (if available) to manage the proofing process, and ensure they are copied into your email to the journal.

Kind regards,

Lianne Parkhouse
Royal Society Open Science
openscience@royalsociety.org

on behalf of the Associate Editor, and Professor Kevin Padian (Subject Editor)
openscience@royalsociety.org

Associate Editor Comments to Author:

Congratulations on the acceptance of this piece of work - we're grateful for your support of Royal Society Open Science and look forward to further submissions to our journal in future.

Reviewer comments to Author:

Reviewer: 2
Comments to the Author(s)

I believe the authors did a nice job at addressing my concerns with the manuscript in a thoughtful manner. I appreciate the clarity provided on some of the methods that I was confused on, and their willingness to make the appropriate changes. I think however that it is a good idea to post all relevant data and scripts in an easy to access format. I commend the authors for their cool and interesting piece of work and their contribution to African and mammalian evolutionary ecology.

Appendix A

Reviewer: 1

The manuscript presents an indirect assessment of community assembly processes based on morphometric analyses of shrews in the Congo Basin. The writing is clear and concise; the methods are sound; and the interpretations are congruent with the results. I have two concerns (see two paragraphs below), each of which the authors should be able to address.

The authors claim to include a table containing “additional details” (see Fig. 1 caption) regarding sampling localities. However, I could not find that table, neither in the main file nor in the file containing Supplementary Info. Without those details, it is difficult to determine the distribution of sampling effort between CLF and NLF. Since 2 of the 6 dots in Fig. 1 are south of the Congo, whereas 4 are north, I suspect that the sampling was uneven. In addition, from the map it looks like the geographic extent of sampling localities was greater for NLF than CLF. If so, it would be useful for the authors to explicitly address the concern that differences in sampling intensity and broader geographic (less spatially restricted) sampling in NLF could themselves reflect as differences in shrew community structure.

> We apologize for not including the additional details. We now included a table with the number of sampling sites per ecoregion and per locality in the manuscript.

We added the following to the discussion: Note that differences in geographic extent and sampling effort between ecoregions can affect richness of the regional species pools. Because of the lower number of sampling sites and individuals captured in the CLF (15 sites and 523 individuals) compared to the NLF (21 sites and 750 individuals), a potential bias could result in an underestimation of the richness and functional volume of the CLF species pool compared to that in the NLF. Despite this potential bias, the NLF species pool contained only 1 more species than the CLF species pool. A priori, we expected the NLF species pool to be more species rich compared to that of the CLF due to the empirical species richness anomaly associated with historical biogeography [12-14]. Consequently, we have little evidence that unequal sampling efforts or sample sizes between the two regions created a bias that would affect the conclusions of this study. Importantly, only analyses based on the total species pool included data that could bias results, as other analyses used a community as the focal unit or were restricted to a single region.

The authors base the justification for a particular null model on a claim that “community assembly transpires in a hierarchical manner, where species first pass through an abiotic filter and then a biotic filter...” (pg. 7), which is incorrect. There is no necessary order to abiotic and biotic filtering in community assembly. For example, imagine a shrew species that arrives to a new site during the wet season, but cannot survive there owing to an endemic disease organism, so its population fails to persist, even though it would also have failed during the dry season that presents conditions outside of the shrew’s fundamental niche, had it persisted through the wet season to the dry. The logic in this paragraph needs to be fixed.

> We have edited the conceptual basis of this hierarchical approach to separate dispersal filters (historical contingency, dispersal barriers) from environmental filters (both biotic and abiotic filters which act in concert in local habitats). This conceptual arrangement better reflects the hierarchical spatial effects of assembly of shrew communities in the Congo.

Line 89 – edit “we only used sessions were restricted to” to improve clarity.

> We changed this to “we only used sessions that were restricted to a single season in order to avoid use of data from transition periods between seasons”.

Line 176 – replace “indicated” with “indicate”.

> *We corrected this.*

Reviewer: 2

This is an interesting study on a fascinating yet under-explored fauna and region of the world. I commend the authors for gathering all of these data, however the location of archived samples and the methods used to identify these poorly studied species need to be addressed. I take some issues with the interpretation of the results, and the strength of the statements in the discussion. The largest concern is the discrepancy between the results that say community ecological volume is increasing with species richness, and the interpretation that “species became more tightly packed in the functional volume rather than expanding the size of the functional volume.” The results seem to be in sharp contrast to this statement. Also, the strength of the conclusion that environment is the cause of the differences between communities is less than the authors make it out to be.

Environmental filtering is one of the alternative explanations, but not the definitive reason. I would also like to see more details in the methods. I hope my comments can be useful to the authors.

Abstract: Lines 23-24: Confusing statement about central and north, perhaps “number of species is lower in the central ecoregion than the northern ecoregion”.

> *We changed “less” to “lower”*

Line 25: “...these climatic variations...” Which climatic variations?

> *We changed “these” to “past”*

Line 27: Environmental factors are not “stronger” in some places than others. Their effects on species richness may be.

> *We changed “which are stronger” to “these constraints are stronger”*

Line 28-30: Sentence that begins “An effect of environmental filtering...” is confusing. It can be hard to fit so much into a limited abstract, but this sentence needs to be re-worded to be understood.

> *We revised this sentence to enhance clarity.*

Introduction: Lines 43-44: I’m not sure about “ecological time-scales”, just using evolutionary time scale here may be more clear.

> *Both ecological and evolutionary processes might be at play, therefore both time frames are relevant.*

Lines 44-45: I don’t know if Hubble said that “community assembly can be a largely neutral process”. I think he modeled assembly as a neutral process and found some theoretical (and a little bit of empirical) support for this model.

> *This sentence has been edited to clarify the relationship between neutral models and equivalence of demographic processes.*

Line 45: Enhance mechanistic understanding of what?

> *We added “of the processes driving community assembly”.*

Line 46-51. It is interesting that both hypotheses, niche expansion and niche packing, could both be driven by specialization. I am not saying that this is what the references intended to say, but it makes me think that limiting similarity and increased packing by finer specialization could be two sides to the same idea, however niche expansion, I am guessing, would require specialization on a resource not previously in the niche volume, and niche packing would require specialization WITHIN the volume, though that may be hard to disentangle. It could be useful to clarify these ideas here and perhaps later in the manuscript.

> *In the context of our manuscript, specialization refers to the reduction of niche space by a species as a means to reduce niche overlap. A species that when added to a community, would expand the niche of that community, is not necessarily specialized to a (new) resource; its resource use may still overlap with that of other species.*

Methods: In general, more information is needed in the methods. One of the biggest things that is missing is how species were delimited. I believe that shrews can be very difficult to tell apart, and I'm sure the shrews of the DRC are likely understudied, and these regions likely include cryptic species. Are these specimens archived? Do add some information of how the authors went from shrews in a bucket to samples with accurate species identifications.

> *We added "Species were identified based on external morphology and cranio-dental characteristics. In addition, species assignments were confirmed for several specimens of each species by molecular analysis (16s rRNA)." as well as references to a recently published data paper that includes more detail.*

Line 85: Tell us more about where Kisangani is. So far, the authors have only mentioned the Congo Basin, not the DRC.

> *We added ", located in the centre of the DR Congo"*

Line 88-90: Wouldn't trapping in multiple seasons potentially yield more species? Since the reader doesn't know how you are going to partition your data for analysis, this information on Yoko is confusing here. It may be better to put it in the next section.

> *We moved the selection of Yoko sites to the next paragraph and we made clear that Each sampling site is considered a shrew community, which means our dataset contains 15 communities of the CLF and 21 communities of the NLF (Table 1).*

Line 104: This is not a complete sentence. Rephrase please.

> *We changed this to "Our selection of ecomorphological traits..."*

Line 108-110: Please give more information of why you did this "size correction" and cite some sources.

> *We used relative measures to characterize function because most morphometric traits were strongly correlated with body size, and functional diversity metrics generally decrease as the strength of positive correlations between traits increases. We added the latter statement in the text and provided a reference.*

Line 114-115: Please state what vegan command you used, number of axes, number of iterations, etc. Also, why was NMDS used rather than some other ordination method? Correct me if I'm wrong, but I don't think RSOS has strict page limits. An extra sentence or two explaining your choice of methods will go a long way in clarifying your results to the reader.

> *We replaced this sentence with "The position of shrew species in two-dimensional trait space was visualized by Non-metric Multidimensional Scaling (NMDS) using the metaMDS function of the R package vegan [30]. We used Gower distances (vegdist function in vegan) to represent pairwise functional distances between species."*

Line 117: Cite R package, and give command and relevant settings.

> *We added "Both measures were calculated for each community in R [32]: we used the dbFD function of the FD package [33] to calculate FRic and the mean Gower distance from each species in the community to its nearest neighbour in multivariate trait space to determine MNND [34]."*

Line 118: Was this convex hull calculated from raw variables, or NMDS scores that were generated in the previous section? Or was some other ordination method used here? This is very important. How was the data partitioned and how many convex hulls are there? 2, one per “region”, or 1 per community (6)?

> *The NMDS was only used to visualize species in trait space, as explained above. We included “Both measures were calculated for each community” to specify that each community has its own MNND and FRic.*

Line 119-120: How was MNND and Gower distance calculated?

> *MNND is the mean distance from each species to its nearest neighbour, as explained above. Gower distance is calculated using the vegdist function in vegan, as explained above.*

Line 128-129: I thought that the “regional species pool” method is what was done in the previous section. It is not clear how the null is different from the alternative. See method comments above.

> *To overcome this confusion, we first characterize null models before we explain why we used two different definitions of the species pool.*

Line 131: Careful with “adapted” here. Isn’t a neutral theory being tested? The total pool just contains all the species in the lowland rainforests.

> *We changed “adapted to” to “able to survive”.*

Line 134: remove “regardless”

> *We removed it.*

Lines 135-137: Explain how you calculated this (which R(?) package), same as above.

> *The calculation of SES was explained on the next line. We specify that all calculations are done in R.*

Lines 142-152: I’m a little confused by this paragraph. This sounds like a justification of the methods that you used in the above paragraph. As far as I can tell, your null models did constrain the traits using empirical trait volume. If I am reading this right, and I may not be, I recommend moving this paragraph to the start of the null model methods section so readers understand your choice of null model evaluation prior to describing the exact methods.

> *The null models explained in the first paragraph randomly pick species from the species pool which means they do not constrain the choice of species based on their traits. The appropriate null model for nearest neighbour distance constrains the species pool by the empirical trait volume of the community. We added the latter statement to the manuscript.*

Results: Line 160: Provide some numbers here, and explain “significantly larger slope” (i.e. effect)

> *We added “(i.e. significant interaction between species richness and ecoregion the ANCOVA, Fig. 2a)” to this sentence. P-values of the interaction between species richness and ecoregion are stated in figure 2. For the sake of clarity, we also moved the reference to figure 2 to this sentence.*

Line 161: Are these slopes “significantly different” as in line 160? This will affect your statement here.

> *No, there is no interaction between species richness and ecoregion (Fig. 2b). We rephrased the sentence: “MNND decreased with increasing species richness in each ecoregion. Slopes did not differ between ecoregions, but higher MNNDs were found in the NLF than in the CLF”.*

Line 167-169: Citation needed.

> *We included a reference to Swenson and Weiser (2014).*

Discussion: Line 202-203: I think you mean “If ecomorphological traits can effectively delimit niche occupancy,..” I don’t think species packing is determined by ecomorphology, and, since you

estimated niche volume FROM ecomorphological traits, then it is inevitable that your volumes are based on these traits.

> *We rewrote this to “When species packing and niche volume...”*

Lines 205-207: I am confused by this interpretation. It seems that functional volume increases with species richness (Fig. 1A). This is a volume-based metric calculated with the FRic method. If FRic increases with species richness, this means that ecomorphological volume is increasing, and therefore this supports the hypothesis of limiting similarity and NOT the hypothesis of niche packing in the same volume. This is the exact opposite of what is said on line 206-207, where you say that functional volume DOES NOT increase with species richness, despite reporting that it does.

> *We have edited this text to remove the conflicting statement which conflated two sets of results.*

The empirical relationships show that with increasing richness, functional volume increases and MNND decreases; so as species richness increases both phenomena occur (more volume and “functional neighbours” are closer). By chance, these patterns are expected to occur with increasing species richness. Consequently, we conducted analyses using null models to determine if increases in functional volume or decreases in mean nearest neighbour distances (MNND) were different than expected by chance. The null model results showed that functional volume increased less than expected by chance in the CLF (but not the NLF) and that MNND decreased no differently than expected by chance.

Consequently, functional volume can increase with richness (Figure 2A) AND at the same time still be less than expected based on a random draw of species from the total or regional species pools (Figure 3A).

Line 208-210: I do not know how environmental factors come into play here, as they, as far as I can tell from the methods, were never tested. You describe the history of the region in the introduction, and then state that the environments north and south of the river are very similar. What I think you mean is that historical factors MAY explain the differences in the communities, not environmental factors DO explain.

> *We have edited the text to clarify that functional richness is constrained in local communities, and that we are hypothesizing that these constraints are environmental, as null model analyses show that functional richness is less than expected due to chance in the CLF based on both total and regional species pools.*

Line 214: Need to describe what “limited” means here. Limited in richness? In morphological disparity?

> *Indeed, limited in richness. We added this in the manuscript.*

Line 230: I don't know if they are “likely limited” as this is just one of many alternative hypotheses. “May be limited...” seems more appropriate.

> *We changed this.*

Lines 235-240: These results are consistent with the hypothesis of limiting similarity. See comments above. Community ecological volume increases with species richness (Figure 1A), and the rare species in the CLF are outside the central morphospace. Figure 5 is a nice example of this. However, aren't the axes and distances in NMDS meaningless? As in the distances represented in NMDS are not “true” distances? A PCA/PCoA plot of this would be interesting to see as well, but maybe not required.

> *We do not believe that these results are consistent with limiting similarity. To support limiting similarity, all species would need to be some minimum distance from each other in functional space.*

As shown in Figure 5, many of the most abundant and frequent species occur very close to each other in functional space. That rare or infrequently occurring shrews in the CLF are on the periphery of functional space may indicate that these shrews simply occupy niches that do not support large populations or that cannot support populations in many locations. Importantly, shrews do not only compete with other shrews, species on the periphery of functional space may be in competition with non-shrews more so than shrews in the center of this functional space.

We selected NMDS because it collapses as much variation as possible into fewer dimensions and can represent the relative positions of species faithfully with acceptable levels of stress. The benefit of NMDS is that it maximizes the variation associated with the desired number of axes, whereas PCA or PCoA would faithfully represent relative positions for only a portion of the variation in the data, with remaining variation existing on axis not used in the visualization.

Line 267-269: ...or because the resources that they are specialized in are less abundant in the habitat, or they are harder to catch in pitfalls.

> We incorporated the reviewer's suggestion.

Line 285: This is very interesting, but I don't think your results make this definitive. "Our results suggest that the biogeographic history of the basin may have left an imprint on...."

> We made the suggested change.

Line 289-290: Isn't the expectation that specialists are in lower numbers than "generalists" found in the center of morphospace? Also I don't think "enjoy" is the best word here. We don't really know what, or if, shrews enjoy.

> Based on competition theory, it can be expected that species that are further from other species in morphological space are less likely to be affected by competitive interactions and therefore are able to attain higher abundance. We changed "enjoy" to "exhibit".

Line 292: consistent with, not necessarily "caused by"

> We changed this.

Line 326: Is the editor of this volume the same as the second co-author? The names are spelled differently

> We corrected the spelling error.

Reviewer: 3

The paper examines coexistence of shrew species in parts of the Congo Basin to explain why one portion of the basin harbors more species than a different portion. The approach is to examine niche space of the different species. The study addresses an important question in species diversity and uses a somewhat interesting approach. The paper is quite readable. My comments pertain to a some of the assumptions, and whether the findings are conclusive or not. Mostly they require clarification for a fairly naïve reader.

1) I am not a shrew biologist, and thus I was a bit surprised to learn, in the discussion, section that all of the species are generalist feeders. Therefore, diet plays no part in niche. This is fine. However, it should have been mentioned in the introduction. Indeed, after reading the discussion, it is not clear to me how you concluded that (line 76) 'competition will be intense'. It seems to me that generalists can switch prey items easily (?).

> In the introduction we mention that shrews generally have very similar ecological niches, this "enhances the likelihood that competition will be intense", which is true from a theoretical point of

view, but by no means is an empirical necessity, depending on the extent to which common resources are limiting. In the second paragraph of the discussion we now added that our findings contradict theoretical expectations and suggest alternative explanations.

2) To me, niche is more than ecomorphological traits. The morphological aspect is clear. However, it would help readers to expand a bit on the ecological aspect. Again, reading the paper alone, it is not clear to me how these species separate among themselves ecologically.

> We added a paragraph in the methods section that explains our choice of ecomorphological traits and emphasize in the discussion that competition is a process that only manifests when resources are limiting. Indeed, for non-random ecomorphological patterns to emerge, competition must be intense and lead to local extirpation, the dominant mechanism affecting species assembly, and characteristic of most species in the community.

3) I assume that quantifying niche packing and filling of functional space statistically is accepted. However, it might help readers to describe the extent of resources in the two portions of the study area. Perhaps, I am too wedded to Geographical Ecology, by Robert MacArthur, by niche packing and filling to me is the broken-stick model and length of the stick. I did not get a feel for available resources by 'multivariate convex hull volume'.

> The broken stick model is one of many models linked to the niches of species. In essence, it characterizes how abundances (energy) are apportioned among species in a community (i.e., relates to rank abundance distributions, not ecomorphological patterns). Such rank abundance models do not address how species are dispersed in functional space. The assumption of all of these models is that competition is the dominant factor moulding the structure of the communities (see previous comment) and that the models deal with the allocation of energy or biomass or number of individuals among species based on subdivision of all resources (and that resources of all types are homogeneous in distribution). As said at the end of the second paragraph of the discussion, we could not calculate niche overlap.

4) The Results section reads well. But there seems to be a large leap of logic from the Results to the last paragraph of the Discussion, stressing the distinctive imprint of Holocene climate. I assume the imprint is in the species composition and ecomorphological traits, by I suspect that naïve readers will miss the connection, too.

> We rewrote the last two paragraphs so that they clearly list the evidence that led us to believe in the imprint of Holocene climate.

Editor

> We used data from published sources, therefore no field study was required for this manuscript.

- Data accessibility

- Competing interests

- Authors' contributions

- Acknowledgements

- Funding statement
